# The Effect of Different Preoperative Depilation Ways on the Healing of Wounded Skin in Mice

**DOI:** 10.3390/ani12050581

**Published:** 2022-02-25

**Authors:** Xinyi He, Lintao Jia, Xiao Zhang

**Affiliations:** Department of Biochemistry and Molecular Biology, Fourth Military Medical University, 169 West Changle Road, Xi’an 710032, China; chellhe@yeah.net (X.H.); jialth@fmmu.edu.cn (L.J.)

**Keywords:** hair removal, surgery, C57BL/6J mice, wound healing, sodium sulfide, electric shaving

## Abstract

**Simple Summary:**

An increasing number of animals, including pets, may undergo surgery due to diseases or cesarean section nowadays. To reduce surgical site infection, hair removal is a necessary step before the surgical procedure because of the dense hair layer on the skin. However, previous studies showed that inappropriate hair removal methods might even increase surgical site infection rates. Although there are many commonly used preoperative depilation methods, it still has no detailed, unified selection criteria for animal preoperative hair removal. Therefore, we intend to provide a scientific and practical reference for veterinarians. To explore more specific details on whether depilation affects the condition of a surgical site, we established a skin wound model after the depilation step and then compared four commonly used hair-removal methods through morphological assessment and histopathological analysis. Ultimately, we concluded that the electric shaving method is the best method for preoperative depilation, followed by the depilatory cream method, and the sodium sulfide aqueous solution depilation method is the worst. We hope that the results of this study can provide useful reference points for veterinarians and researchers and help refine surgical procedures and maintain animal welfare.

**Abstract:**

Hair removal is necessary before operating on animals with dense hair layers. To provide an appropriate hair removal method and maintain animal welfare, we introduced four commonly used depilation methods—namely, scissors shearing, electric shaving, depilatory cream, and sodium sulfide, and made systematic comparisons, instead of only examining one or two methods, as reported in the past. To further assess convenience and possible effects on skin wound healing, we performed a skin trauma model after depilation in C57BL/6J mice and recorded wound healing time. Meanwhile, the skin tissues around the wound were stained with H&E and Masson. The results showed that the wound contraction rate of the sodium sulfide group was significantly lower than other groups at different points in time. Furthermore, depilatory cream and sodium sulfide methods could induce a topical inflammatory response on the third day after the operation and delay the regeneration of collagen fibers. We concluded that sodium sulfide depilatory has a significant negative effect on wound healing. Depilatory cream is gentler, with mild skin irritation and symptoms of inflammation. The electric shaving method is more convenient and safer, and thus could be the best choice for preoperative depilation.

## 1. Introduction

As an essential part of human life, more and more animals have been accepted to become our beloved companions and family members, especially rodents. Although they live longer with good care, they are still likely to suffer from diseases in life, as well as early spay/neuter of pets, cesarean section, etc. [1,2,3]. Many domestic pets have a common feature—namely, that their skin is covered with a dense hair layer, which makes hair removal necessary before surgery. Despite previous studies suggesting that the best practice is to refrain from hair removal, the hair-removal step is necessary for some operations because it can expose the skin, which is convenient for operation and prevents the introduction of bacteria to reduce the risk of surgical site infections (SSIs) [4,5,6].

Most previous studies were confined to preventing SSIs through postoperative management, such as wound dressings or negative pressure therapy [7]. However, the preoperative skin preparation can also affect the condition of the skin, which may result in SSI, or even influence the progression of the disease. The preoperative skin preparation includes hair removal and disinfection of the surgical sites. It has been demonstrated that different disinfection methods indicated no significant differences, irrespective of depilatory agents or electric clippers depilation methods used [8,9]. However, in previous experiments, we have found fortuitously that mice depilated by sodium sulfide aqueous solution and scissors hair cutting methods presented some problems in the stages of wound healing. Despite those findings, there were no studies that assessed the difference between these commonly used preoperative hair removal methods in animals.

Hair is a complex biological structure that includes the dermal sheath, outer root sheath, and root sheath [10]. Depilation can remove the section outside the hair follicle, which is made up of a protective cuticle layer and inner cortex. The hair removal methods commonly used can be roughly divided into two categories—chemical and physical methods [11]. Chemical depilatories include depilatory cream and sulfide aqueous solution, as well as physical techniques, including scissors shearing, electric shaving, waxing, lasers, etc. [12].

The physical depilation methods, such as shearing and shaving, exert their function by cutting off the hair outside the skin directly but might damage the skin randomly. Some studies suggested that using clippers or depilatory cream probably results in few SSIs, compared with shaving using a razor [13,14,15]. However, some studies also indicated that both electric razor clipping and depilatory agents are safe in rats and mice [9,16]. Chemical depilatory can break the disulfide linkages on the keratin protein backbone by enhancing aqueous solubility, leading to hair removal [17]. Both thioglycolic acid and sodium sulfide, the chemical composition of depilatory cream, can achieve such an effect [18]. It has been demonstrated that hair removal by depilatory does not affect survival in rodent experimental flaps [19]. Meanwhile, sodium sulfide aqueous solutions were also reported to remove hairs [20]. Some studies have reported that preoperative hair removal by depilatory cream could achieve a higher asepsis score on the first day after surgery [11]. Although chemical depilatory can dissolve hair thoroughly, it may also stimulate the skin and cause an inflammatory reaction. For example, some studies confirmed that tumor-promoting inflammation closely resembles the inflammatory processes typically found during development, immunity, and tissue repair [21]. As reported, trichoblastoma was, by far, the most common skin neoplasm in pet rabbits [2]. Therefore, considering the reduction in impact on the wound and the perspectives of animal welfare, the potential side effect on skin condition and wound healing should be a principal factor in selecting the depilation method. Moreover, we should consider other factors, such as practicability, convenience, cost, etc. [22]. As described above, most previous studies have only examined one or compared two commonly used hair removal methods, and there are no detailed, unified criteria for animal preoperative hair removal selection.

It is necessary to explore appropriate hair removal methods and provide a scientific and practical reference for veterinarians and biomedical researchers. Thus, the aim of this study was to evaluate the effects of four commonly used hair removal methods on skin wound healing and inflammatory response, including scissors shearing, electric razor shaving, commercially available depilatory cream, and sodium sulfide aqueous solution, to provide an appropriate preoperative hair removal method and maintain animal welfare. We hypothesized that the physical depilation methods including shearing and shaving would be the best choice for preoperative depilation.

## 2. Materials and Methods

### 2.1. Animals and Ethics Statement

C57BL/6J mice were purchased from the Laboratory Animal Center of Air Force Medical University (FMMU). They were housed in plastic cages and kept in a regulated environment (22 ± 1 °C), with an artificial 12 h light–dark cycle (lighted from 7:00 a.m. to 7:00 p.m.). This study was conducted under the ARRIVE guidelines [23]. All mouse experiments were performed in compliance with the guidelines approved by the Laboratory Animal Welfare and Ethics Committee of FMMU (Approval No. IACUC-20191005). All animals were restored to good health after the experiments and kept in the animal center until natural death.

### 2.2. Anesthesia

Mice were anesthetized by the dosage of 80 mg/kg ketamine and 10 mg/kg xylazine (H20054748, Hanfeng pharmaceutical company, Xi’an, China; 087K1253, Sigma Aldrich, Darmstadt, Germany) with the intraperitoneal injection [24]. Body weight was measured using the electronic scales (JM-B3002, Chaoze weighing apparatus, Shaoxing, China) and recorded. Then, we returned them to their cage until the righting reflex was lost. After confirmation of appropriate anesthetic depth, the mice were placed on a heating pad and covered their eyes with artificial tears (210303, ZSM, Hangzhou, China). Depilation and surgery were performed in order. After the surgery, we returned them to their cage when the righting reflex recovered.

### 2.3. Depilation and Trauma Model

In total, 32 C57BL/6J male mice (6 weeks old) were randomly divided into 4 groups. Mice were anesthetized as described above. The hair on the abdomen from the xiphoid process to the thigh root was removed using the different depilation methods. The hair around the surgical site was soaked with warm sterile sodium chloride physiological solution (H20103227, BBCA Pharmaceutical, Hefei, China) and cut with surgical scissors (S12005-10, RWD, Shenzhen, China) in the shearing group. The hair was shaved directly by a pet shaver (DDG-S01, Yizu, Hangzhou, China) in the shaving group. In the two chemical depilation groups, the hair around the surgical site was evenly covered with depilatory cream (20070747, Veet, Massy Cedex, France) or 8% sodium sulfide aqueous solution (200108, Tianli Chemical Reagent, Tianjin, China). The duration of depilatory cream exposure was about 60 s, and the exposure time of sodium sulfide aqueous solution was about 120 s. Then, the hair was cleaned by saline solution when the hair was dissolved. The total time depilation taken was also recorded.

After depilation, the skin of the surgical site was sterilized with iodophor. A skin trauma model was built on the central location of the depilation area by cutting the 1 cm diameter full-thickness skin with surgical scissors. Sterilization of surgical instruments was performed in an autoclave at 121 °C for 20 min in a sealed autoclave bag. Then, we pressed the wound with sterile swabs to stop the bleeding and took photos of surgical sites. After the operation, we put them on the heating pad to keep them warm. When the righting reflex was confirmed, 0.1 mg/kg buprenorphine (Y0001108, MREDA, Beijing, China) was injected subcutaneously with 28.5-gauge insulin needles [25]. Subsequently, they were returned to their cages.

### 2.4. Calculation of Wound Contraction Rate and Wound Healing Time

Mice were anesthetized by the dosage of 50 mg/kg sodium pentobarbital (P-010, Sigma-Aldrich, Darmstadt, Germany), with intraperitoneal injection on the 1st, 3rd, 5th, 7th, 9th, and 11th day after the operation [26]. Then, wound evaluation was performed including color, swelling degree, scab formation, granulation growth, and wound contraction rate. In addition, the body weight of each mouse was measured by electronic scales (JM-B3002, Chaoze weighing apparatus, Shaoxing, China). Body weight changes were expressed as percentages, and weights on the operation day were set for 100%. The wounded area was photographed and statistically analyzed at each time point (Figure 1A).

ImageJ software was used to measure the wounded area. They were measured at different time points, and the wound contraction rate was calculated (R wound contraction rate = (1 − S wound area/S initial wound area) × 100%). When the numeric value of contraction rate exceeded 90%, it was judged as a healed wound, providing the wound healing time.

### 2.5. Histopathology

Two mice were randomly selected from each group on the 3rd, 7th, and 11th day after the operation (Figure 1A). About 3–5 mm full-thickness skin was taken out along the wounded site after anesthesia and then fixed with 4% paraformaldehyde (G1101, Servicebio, Wuhan, China), paraffin-embedded, and sectioned. The anesthesia and analgesia procedures were performed as described above. H&E and Masson’s trichrome staining were carried out for histopathology observation. The pathological changes in skin wounds were observed under a microscope and photographed for analysis (BX51, Olympus, Tokyo, Japan).

The H&E-stained sections were used to evaluate inflammatory cell infiltration and hair follicles regrowth. The number of inflammatory cells of each treatment of one animal was manually counted from the digital images of photomicrographs that were taken from four representative areas of each tissue section at a fixed magnification of 40×. Additionally, they were chosen to count the number of hair follicles from the wound edge to 3 mm away at a fixed magnification of 5×. The collagen volume fraction (CVF) (CVF = collagen area/total tissue area) was calculated to measure the content of collagen fibers using the Masson’s trichrome-stained sections. The values of collagen and total tissues were measured by ImageJ software, at 10× magnification.

### 2.6. Statistical Analysis

Statistical analyses were performed using GraphPad Prism (version 8.0, GraphPad Inc., La Jolla, CA, USA). Data are expressed as the mean ± SD. The comparisons between four groups were performed by one-way ANOVA analysis of variance. Differences were considered statistically significant when * *p* < 0.05; ** *p* < 0.01; *** *p* < 0.001.

## 3. Results

### 3.1. Sodium Sulfide Tends to Cause the Symptoms of Inflammation around the Wounded Area

Since the presence of different kinds of depilation methods, we firstly assessed the rapidity of hair removal in mice. Obviously, the total hair removal time using different methods was significantly different. Shaving was the best way to save time, while shearing took the longest time. The other two chemical methods were performed in medium time (Figure 1B). Then, the skin trauma model was established to investigate the effects of different hair removal methods on wound healing and skin regeneration. There was no significant difference in the weight change between the four groups after the operation (Figure 1C). However, compared with physical depilation groups, wound sections in chemical depilation groups showed symptoms of surgical site infection. The skin around the wound was red and accompanied by yellow exudate, especially in the sodium sulfide group (Figure 1D). In addition, erythema and desquamate symptoms around the wound in chemical depilation groups were observed.

### 3.2. Sodium Sulfide Aqueous Solution Depilation Prolongs the Healing Time

Meanwhile, we measured the wound contraction rate and collected the data of healing time in all groups. On the 1st, 3rd, 5th, 7th, 9th, and 11th day after the operation; there were no significant differences in wound contraction rate among the three groups including shearing, shaving, and depilatory cream. However, the contraction rate in the sodium sulfide group was found to be dramatically lower than that in other groups on the 1st, 3rd, 5th, and 7th day after injury. From the 9th day, the difference in wound contraction rate between the sodium sulfide group and other groups narrowed, but there was still a significant difference (Figure 2A). In addition, we found that the healing time of the sodium sulfide group was markedly extended. The statistical analyses of wound healing time also confirmed that the healing time in the sodium sulfide group was significantly longer than the healing time in other groups (Figure 2B).

### 3.3. Chemical Depilation Enhances the Inflammatory Response of Wounded Area

H&E staining was further performed to estimate the surgical site inflammation. Inflammatory cells, such as neutrophils and mononuclear cells, were found in the wounded area on the 3rd day after the operation in all the groups (Figure 3A). Furthermore, there were significantly more inflammatory cells in the sodium sulfide group than in physical depilation groups, and the depilatory cream also caused a mild inflammatory response (Figure 3B). In addition, we examined hair follicles during wound healing. On the 3rd day after the injury, we found that the number of hair follicles was dramatically decreased in the dermis around the wounded area. From the 7th day, the hair follicle began to regenerate and arrange in a compact and orderly manner (Figure 3C). However, on the 11th day, the number of hair follicles at the wound healing areas in the sodium sulfide group was less than that in other groups (Figure 3D).

### 3.4. Chemical Depilation Decreases the Content and Delays Regeneration of Collagen Fibers

Meanwhile, collagen fiber was measured to estimate the wound healing state through Masson’s trichrome staining, and collagen fiber was stained with blue or bright blue. Compared with the sodium sulfide group, more collagen fiber and a shorter collagen tissue gap were observed in the other three groups on the 3rd day after the operation (Figure 4A). Additionally, the content of collagen fibers in the depilatory cream group was slightly less than the two physical depilatory groups (Figure 4B). Although the differences between the groups remained significant after 7 days, the differences had narrowed substantially (Figure 4C). Furthermore, collagen fiber contents were similar in the three groups except for the sodium sulfide group on the 11th day. There was still a significant difference between the sodium sulfide group and the other three groups (Figure 4D).

## 4. Discussion

As the part that directly contacts the external environment, the skin plays a vital role in the immune system. Surgery and trauma cause skin wounds, leading to many interactive physiological responses including hemostasis, inflammation, re-epithelization, and remodeling that occur when the skin is damaged [27]. Along with these reactions, the cutaneous wound repairs rapidly and efficiently through a series of processes, such as cell proliferation, angiogenesis, collagen regeneration, and scar tissue formation [28,29]. As a preoperative procedure for surgery, hair removal should not cause additional damage to the skin or even affect the wound healing process. Unfortunately, in actual surgical procedures, due to the limitations of various hair removal methods, it will more or less affect the skin condition, thereby affecting the tissue repair process, or even resulting in more severe disease.

As we know, wound healing and tissue repair is an intricate biological process that involves the repair of cellular damage and maintenance of tissue integrity. However, several studies conclude that cascades involved in wound healing and tissue regeneration highly overlap with cancer-causing pathways. Usually, wound healing events include the release of a number of cytokines to accomplish post-trauma restoration [30]. For example, as a fundamental mediator in inflammation, wound healing, and tissue regeneration, IL-22 also plays a pivotal role in the instigation of various cancers due to its pro-inflammatory and tissue repairing activity [30]. Li et al. assessed the interaction between skin inflammation and wound healing, and their data showed that using exogenous Smad7 below an oncogenic level can alleviate skin inflammation and wound healing defects associated with excessive activation of TGF-β and NF-κB [31]. Therefore, when we perform animal surgery or conduct research on tumors, trauma, and other diseases, we should pay attention to the impact of hair removal methods on wound repair.

In previous studies, skin trauma models were often used to evaluate the possible effects of some preoperative procedures on wound healing, such as disinfection and hair removal. As described above, wound healing is a slow and complex process that requires weeks in specific conditions and undergoes five overlapping stages [32]. Any abnormality at these stages will negatively affect skin wound healing. With the extension of healing time, the dynamic system could be caught in a vicious circle of muscle spasm from local blood flow reduction, inflammatory [33,34]. Therefore, wound healing time is an important metric for choosing a hair removal method. In this study, we recorded the healing time by measuring the contraction rate at each time point and assessing the wound condition. As shown in Figure 1D and Figure 2B, these results indicated that chemicals especially sodium sulfide were more likely to cause symptoms of inflammation in the wounded area and prolonged the time of wound healing significantly. Although the data showed no significant difference in wound contraction rate among the other three groups, considering the chemical reagents and the thoroughness of hair removal, we should make a choice according to the experimental requirements. In addition, mice would typically lick the wound after a skin wound. Pemmari et al. explored whether the licking may affect wound healing because carbonic anhydrases (CAs) VI in saliva contributes to tumor cell migration. However, the results demonstrate that CA VI does not play a significant role in skin wound healing and suggest that saliva-derived CA VI is not responsible for the licking-associated improved wound healing in animals [35]. These remind us that we should pay attention to more uncertain factors affecting wound healing.

Then, we sought to evaluate the healing process through morphological observation and pathology examination, such as the soakage of inflammatory cells, as delayed healing is related to inflammatory dysregulation. Furthermore, dysregulation could lead to susceptibility to infection, unsuccessful healing, hypertrophic scarring, and keloid formation [36]. Then, it will turn to chronic inflammation characterized by abundant neutrophil infiltration, and healing proceeds only after inflammation is controlled [37]. As shown in Figure 3, we found that the number of inflammatory cell infiltration induced by chemical hair removal was significantly greater than that caused by physical hair removal. This is consistent with the results of a previous study indicating that the main composition of depilatory cream triggered the depilatory-induced, lipid-mediated skin inflammatory responses [38]. As reported, a dense and compact arrangement of hair follicles and sweat glands often indicates good wound healing [39]. The data suggested that sodium sulfide made a significant impact on healing, compared with depilatory cream and physical depilation methods. Furthermore, it has been reported recently that compared with shaving methods, commercially available depilatory creams caused a transient minor inflammatory response of the skin and increased the levels of cytokines that might subsequently affect hair growth but increased the number of hair follicles [40]. Coincidentally, this can also be observed in Figure 3D in the present study. Furthermore, depilatory cream can slightly affect skin healing and changes in collagen arrangement through changing the arrangement and structure of cornified cells [41]. Therefore, as one of the criteria to measure the wound healing process, collagen regeneration is another important reference index.

To further evaluate the repair process, Masson’s trichrome staining was performed to explore the collagen change at each time point. As shown in Figure 4, the content of collagen fibers markedly reduced in the sodium sulfide depilation group but slightly affected in the other three groups, while more collagen formation early in the wound healing process could accelerate the repair of the damage site and make the scar stronger [42]. Although the content of collagen fibers gradually increased in wound healing, collagen fibers were more tightly arranged in the two physical depilation groups. This means that the chemical depilatory may affect the wound healing process through the content of collagen fibers regeneration. In addition, the time of collagen fibers regeneration was obviously prolonged in chemical depilatory groups, especially in the sodium sulfide group. However, we did not verify additional factors that can reflect the wound healing process, such as the number of fibroblasts and the type III collagen content. Therefore, in preoperative preparation, we should use an appropriate depilation method that has the least impact on the animal skin condition.

In the present study, we found that scissors shearing was not the best hair removal method. Although it removes hairs without touching and stimulating the skin, it can easily hurt the skin without skilled operation and cannot remove hair thoroughly. Surgical site infection was easily found in this group, which is consistent with previous studies [13,16]. Furthermore, we should not use sodium sulfide when conducting wound healing or dermatological research, because it may introduce more factors and affect research conclusions. The electric shaving method is better for more preoperative hair removal to reduce the impact on wound repair, thus could be the best choice for preoperative depilation based on its convenience and safety. Depilatory cream is also a suitable method due to its minimal influence. In a future study, we intend to explore the mechanism of sodium sulfide influences in wound healing and whether the effects differ between sexes. Overall, we hope these data can provide useful reference points for veterinarians and biomedical researchers, helping to refine surgical procedures and maintain animal welfare.

## Figures and Tables

**Figure 1 animals-12-00581-f001:**
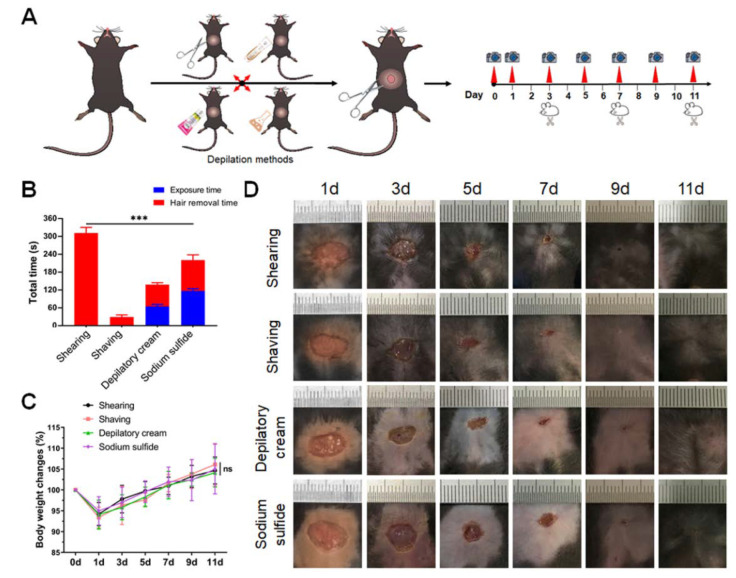
Sodium sulfide caused the symptoms of inflammation around the wounded area. The setup and morphological observation of skin wound model in C57BL/6J mice: (**A**) schematic of skin trauma model with different hair removal methods; (**B**) depilation time of different methods; (**C**) body weight changes in mice on 1st, 3rd, 5th, 7th, 9th, and 11th day after injury; (**D**) representative images of wound healing at each time point. Data are shown as mean ± SD. *** *p* < 0.001.

**Figure 2 animals-12-00581-f002:**
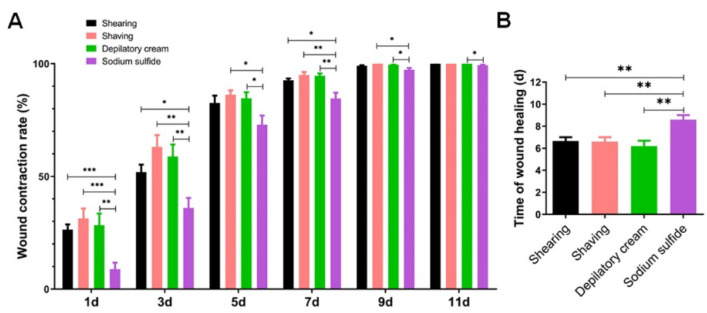
Sodium sulfide reduced the wound contraction rate and extended the healing time: (**A**) the proportion of the wound healing relative to the initial wound area in different groups at each time point; (**B**) time of skin wound healing in different models. Data are shown as mean ± SD. * *p* < 0.05, ** *p* < 0.01 and *** *p* < 0.001.

**Figure 3 animals-12-00581-f003:**
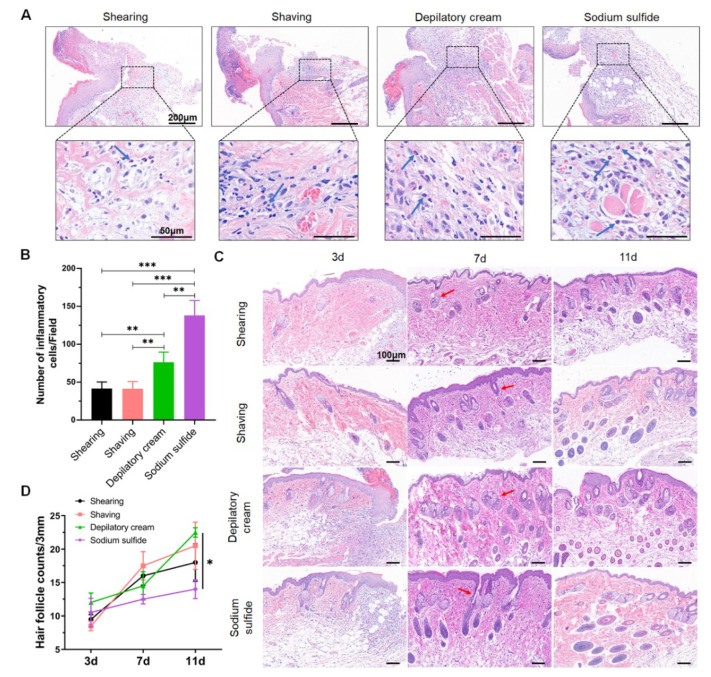
Chemical depilation enhanced the inflammatory response around the wounded area and sodium sulfide delayed skin regeneration: (**A**) H&E staining of the wound sections on 3rd day after injury. Blue arrows showed inflammatory cell infiltration. Scale bars are 200 μm (upper) and 50 μm (bottom); (**B**) quantification of inflammatory cells from randomly four fields around the wounded area; (**C**) representative images of the hair follicles at each time point around the wound. Red arrows showed hair follicles. Scale bar is 100 μm; (**D**) changes in hair follicle numbers during wound healing. Data are shown as mean ± SD. * *p* < 0.05, ** *p* < 0.01 and *** *p* < 0.001.

**Figure 4 animals-12-00581-f004:**
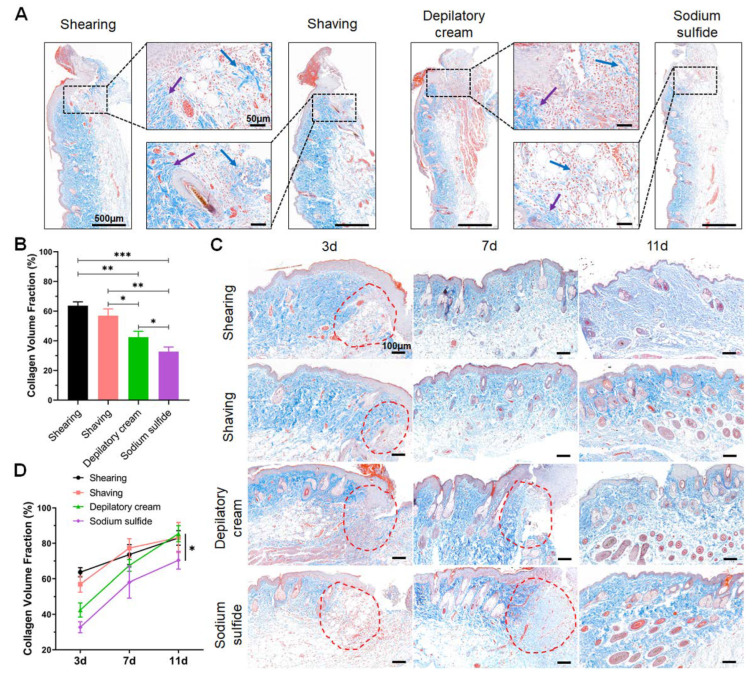
Chemical depilation decreased the content and delayed the regeneration of collagen fibers. Masson’s trichrome staining of the wounded area: (**A**) collagen content of surgical sites 3 days after injury. Arrowheads point to collagen fibers, blue arrows indicate thinner collagen fibers, while purple arrows indicate thick collagen fibers. Scale bars are 500 μm and 50 μm; (**B**) quantification of the collagen fiber around the wounded area on 3rd day; (**C**) representative images of the collagen fiber at each time point. Red circles showed collagen fiber around the wounded area. Scale bar is 100 μm; (**D**) changes in the content of collagen fiber during wound healing. Data are shown as mean ± SD. * *p* < 0.05, ** *p* < 0.01 and *** *p* < 0.001.

## Data Availability

The data presented in this study are available on request from the corresponding author.

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
