# Peer review of "The Effect of Different Preoperative Depilation Ways on the Healing of Wounded Skin in Mice"

_animals, 2022, doi:10.3390/ani12050581_

Round 1
Reviewer 1 Report
Citations
Citations in the text (e.g.Line 70) should name the first author instead of the last author. Please change this throughout the manuscript.
Title:
Title should be improved.
Simple summary:
This refers a bit too much to companion animals, as the main focus is on mice.
Abstract:
I do not fully agree with the statement that there are no studies investigating presurgical depilation. Studies have compared depilation methods in humans since many years. In rodents some parts have been also studied. Please revise and adapt the manuscript in consequence.
Line 37: Why do the authors consider the electric shaving the more convenient method?
Introduction:
The topic of the manuscript is covered, but more related literature could be included to reflect the current state of the topic.
The hypothesis should be included.
Material and Methods:
The reporting does not follows the ARRIVE guidelines. This should be improved.
Details about the husbandry conditions are missing.
Line 100: Please explain the anesthesia in detail.
Line 100: Please specify the sex of the mice.
Line 102-105: Please explain in detail how the hair was removed (e.g. exposure time to the depilation crème, area, …)
Line 106: Do I understand this right? A 1cm skin wound was remained untreated? I consider this is painful for the animals. Did the mice get analgesics?
Line 106: Please explain in detail how the wounds were set (depth…). Usually wound are sutured after surgery. Why did the authors decide for this potentially painful procedure?
Line120: The analysis of only 2 mice at the different time points is not very significant. The n number should be higher, to increase the validity.
Line 121: How are the mice anesthetized? Do they survive this procedure? Otherwise, the authors should state the euthanasia method.
Please add the details about the analysis of the pictures.
Results:
There are some repetitions from the Material and Methods part and conclusions, which belong in the Discussion section. This needs improvement.
All values and SD, as well as the results of the statistical analysis should be given. Some results are stated showing just one representative picture. This has to be improved.
Line 144-145: This is a conclusion and should be moved to the discussion. This should also be amended in the following results part.
Figure 1 A: This belongs in the M&M section.
Figure 2 B: Body weight change should be given in %.
Figure 3 C: The images are from different mice, at least in the depilatory cream group? It would be more consistent if they were from one mouse. The 3d picture of the depilatory cream suggests that the hair removal did not work that well.
Line 159-160: The difference is still significant!
Line 193: If you mention the number and size of follicles, you should provide a quantification. Otherwise, it is a representative scheme prone to subjective evaluation (even the image itself lacks of explanation). Please provide additional data.
Line 201-204: There is no corresponding data shown, only one representative image.
Line 176: To which timepoint are you referring here.
Discussion
I am afraid the authors obviated key literature along the manuscript. It would make the discussion much more smoothly. Even of some literature is old, consider revising more evidence (below some of the available related papers):
Kick, B. L., Gumber, S., Wang, H., Moore, R. H., & Taylor, D. K. (2019). Evaluation of 4 Presurgical Skin Preparation Methods in Mice. Journal of the American Association for Laboratory Animal Science : JAALAS, 58(1), 71–77. https://doi.org/10.30802/AALAS-JAALAS-18-000047
Angel, M. F., Jorysz, M., Schieren, G., Knight, K. R., & O'Brien, B. M. (1992). Hair removal by a depilatory does not affect survival in rodent experimental flaps. Annals of plastic surgery, 29(4), 297–298. https://doi.org/10.1097/00000637-199210000-00003
Adisa, A. O., Lawal, O. O., & Adejuyigbe, O. (2011). Evaluation of two methods of preoperative hair removal and their relationship to postoperative wound infection. The Journal of Infection in Developing Countries, 5(10), 717-722. https://doi.org/10.3855/jidc.1527
Tanner, J., Norrie, P., & Melen, K. (2011). Preoperative hair removal to reduce surgical site infection. The Cochrane database of systematic reviews, (11), CD004122. https://doi.org/10.1002/14651858.CD004122.pub4
Karegoudar, J. S., Prabhakar, P. J., Vijayanath, V., Anitha, M. R., Surpur, R. R., & Patil, V. M. (2012). Shaving Versus Depilation Cream for Pre-operative Skin Preparation. The Indian journal of surgery, 74(4), 294–297. https://doi.org/10.1007/s12262-011-0368-5
Alexander, J. W., Fischer, J. E., Boyajian, M., Palmquist, J., & Morris, M. J. (1983). The influence of hair-removal methods on wound infections. Archives of surgery (Chicago, Ill. : 1960), 118(3), 347–352. https://doi.org/10.1001/archsurg.1983.01390030079013
Kjønniksen, I., Andersen, B. M., Søndenaa, V. G., & Segadal, L. (2002). Preoperative hair removal--a systematic literature review. AORN journal, 75(5), 928–940. https://doi.org/10.1016/s0001-2092(06)61457-9
The discussion has unfortunately some further significant weak points:
1.The findings are not logically explained.
- Low effort was made in the comparison between their findings and the current findings in the research field.
- The implications of the findings for future research and potential applications are barely discussed.
- If more robust data is included, the conclusion would be supported.
- No limitations are discussed.
- No contradictory data are discussed, especially because the omitted related papers.
Line 224-247 : Literature on inflammation and tumor growth is cited, but not related to the findings.
Line 302-303: ?
Reviewer 2 Report
This article looks at different hair removal methods in mice and how they affect surgical wound healing. This is an interesting topic as many researchers and veterinarians are in need of hair removal methods that don't either affect the study or the recovery and healing of the patient. Overall it's a good article but all the results looked at need to be in the methods, and better overarching discussion points were also in the results and need to be extrapolated in the discussion.
Some specifics are below:
Intro line 44-46: It isn't relevant to the study to list the different types of surgeries performed.
Intro line 46: Delete the word while.
Intro line 69: It states, "studies suggest that depilatory is a better choice," but it doesn't specify which depilatory method is a better choice.
Intro line 91-92: The last sentence of the introduction may be better suited in the results or discussion section because it is showing what the findings were from the study.
Section 2.2 mouse model: induction method, need more details on location, how much hair was removed, and contact time for cream and sodium sulfide.
Line 107: Were the surgical scissors sterile?
Line 137-138: It is mentioned that there "was no significant difference in the weight change," but body weight measures were not mentioned in methods/materials section. Include recording weight in methods section.
Line 138-139: The sentence "The wound healing progress was observed and analyzed at different time points after surgery," could be deleted because it is mentioned above that the healing process was observed/analyzed.
Figure 1A could be referenced in the methods too.
Line 172-174: It mentions inflammatory cells were found, but were the inflammatory cells found in all 4 methods of hair removal or which methods were they found in?
Line 176-178: These lines are discussion and statement and should be moved to the discussion section.
Line 179-180: It mentions what indicates good wound healing which should be added to the discussion section to explain the results, and the sentence starts out, "As we know," but where do we know this from? Add to the introduction what indicates good wound healing.
Line 184-186: Indicate in the methods section that follicle regrowth was looked at.
Line 195-198: It is stated collagen fiber was measured, so it needs to be stated in the methods section. Be sure all your methods are listed in the methods section.
Line 225-247: The paragraph is discussing tumor growth, but it is not mentioned as a focus of the study before. It may help the study some, but tumor growth might need to mentioned before.
Line 302-305: Is shearing enough to maintain a sterile surgical site?
Line 312: Instead of genders use animals and sexes.
Reviewer 3 Report
The authors investigated to determine the best method to remove hair for the animal surgery using the experimentally wounded mice. They investigated 4 different methods to remove hair, scissors shearing, electric shaving, depilatory cream, and sodium sulfide, and concluded that the electric shaving was the best. The result may provide veterinarians and biomedical researchers to conduct animal surgery helpful information to remove hair of animals. However, the following ethical concerns should be clarified before acceptance.
- Analgesics were not provided after surgery. The authors should justify not providing analgesics.
- Describe the fate of mice used in this investigation in the Method section.
Round 2
Reviewer 1 Report
The authors have thoroughly revised the manuscript and improved all criticized points.